# Study of Processing Conditions during Enzymatic Hydrolysis of Deer By-Product Tallow for Targeted Changes at the Molecular Level and Properties of Modified Fats

**DOI:** 10.3390/ijms25074002

**Published:** 2024-04-03

**Authors:** Tereza Novotná, Pavel Mokrejš, Jana Pavlačková, Robert Gál

**Affiliations:** 1Department of Food Technology, Faculty of Technology, Tomas Bata University in Zlín, Vavrečkova 5669, 760 01 Zlín, Czech Republic; t2_novotna@utb.cz (T.N.); gal@utb.cz (R.G.); 2Department of Polymer Engineering, Faculty of Technology, Tomas Bata University in Zlín, Vavrečkova 5669, 760 01 Zlín, Czech Republic; 3Department of Fat, Surfactant and Cosmetics Technology, Faculty of Technology, Tomas Bata University in Zlín, Vavrečkova 5669, 760 01 Zlín, Czech Republic; pavlackova@utb.cz

**Keywords:** deer tallow, by-product, lipase, molecular research, processing conditions, hydrolysis, characterization, novel applications, pharmacy, cosmetics matrices

## Abstract

In most cases, the unused by-products of venison, including deer tallow, are disposed of in rendering plants. Deer tallow contains essential fatty acids and can be used to prepare products for everyday food and advanced applications. This work aimed to process deer tallow into hydrolyzed products using microbial lipases. A Taguchi design with three process factors at three levels was used to optimize the processing: amount of water (8, 16, 24%), amount of enzyme (2, 4, 6%), and reaction time (2, 4, 6 h). The conversion of the tallow to hydrolyzed products was expressed by the degree of hydrolysis. The oxidative stability of the prepared products was determined by the peroxide value and the free fatty acids by the acid value; further, color change, textural properties (hardness, spreadability, stickiness, and adhesiveness), and changes at the molecular level were observed by Fourier transform infrared spectroscopy (FTIR). The degree of hydrolysis was 11.8–49.6%; the peroxide value ranged from 12.3 to 29.5 µval/g, and the color change of the samples expressed by the change in the total color difference (∆E*) was 1.9–13.5. The conditions of enzymatic hydrolysis strongly influenced the textural properties: hardness 25–50 N, spreadability 20–40 N/s, and stickiness < 0.06 N. FTIR showed that there are changes at the molecular level manifested by a decrease in ester bonds. Enzymatically hydrolyzed deer tallow is suitable for preparing cosmetics and pharmaceutical matrices.

## 1. Introduction

Most lipids are composed of triacylglycerides (TAGs) made up of three fatty acids (FAs) bound to glycerol. The properties of lipids depend on the type and structural conformations of the fatty acids, which include saturated fatty acids (SFAs) and trans fatty acids (TFAs) [1]. Traditional techniques for obtaining modified fats include hydrogenation, fractionation, and chemical interesterification. Relatively fewer common methods of structuring lipids include enzymatic hydrolysis, esterification, and interesterification [2,3].

During the hydrolysis of fats and oils, triacylglycerols are cleaved into fatty acids and glycerol with water present. Hydrolysis can be conducted by high-pressure steam cleavage, saponification (alkaline hydrolysis), and biotechnological processes using lipases [4]. Biotechnological fat processing has virtually no side effects and is efficient and safe. Microbial lipases are an essential group of biotechnologically valuable enzymes that, in addition to hydrolyzing fats [5], also catalyze other reactions such as esterification, interesterification, acidolysis, alcoholysis, and aminolysis [6,7].

Changes at the molecular level occur during hydrolysis when FAs are cleaved. Parallel to hydrolysis, esterification also occurs, which is the reverse process. The ester bonds are redistributed during the reaction under enzyme or acid catalysis. During the redistribution, the FA is bound to another position of the glycerol [4]. The balance between the direct reaction (hydrolysis) and the reverse reaction (esterification) is controlled by the water content in the reaction mixture. If the mixture has an excess of water, hydrolysis is predominant, while under conditions where water is eliminated, esterification is preferred. If the esterification conditions of the reactants are set correctly, the esterification of fatty acids with glycerol can produce monoacylglycerols, diacylglycerols, and triacylglycerols. Enzymatic hydrolysis and esterification are used in industrial applications to modify fats [8]. By modifying fats, products with desired functional properties such as melting point, solid phase content, and oxidative stability can be obtained [9]. These processes change fats and oils’ functionality and nutritional properties [10]. 

Fundamental chemical analyses of fats include the peroxide value (PV), which characterizes the oxidative stability of fats [11,12]. Then, the acid value (AV) determines the amount of free FA in the fat, and the saponification value determines the amount of alkali needed to saponify a certain amount of fat or oil [13]. The iodine value (IV) indicates fat unsaturation [14]. Other analyses include determining fat color, textural properties, or saponification value (SV) [15]. The color of the fat is primarily determined by the composition of the feed, which is a defining factor for FAs, and this may be reflected in a brighter or duller color [16]. Fourier transform infrared spectroscopy (FTIR) is used to characterize functional groups [17]; the spectrum for lipids is measured in the mid-infrared region (4000–650 cm^−1^) [18]. Hardness, spreadability, adhesiveness, and stickiness describe the mechanical properties of fats. Hardness indicates the functional resistance against deformation of the material, which is essential, e.g., for hardened fats or margarine [19,20,21]. Cone penetrometry is most often used to measure said parameters [22]. 

Fats are essential food components used in producing cosmetics and pharmaceuticals [23]. In 2022, about 22 thousand tons of pork, 6.5 thousand tons of beef, and more than 12 thousand tons of poultry meat were produced in the European Union [24]. With such a high number of animals slaughtered, a significant amount of animal waste, including fat, is produced. The processing of secondary fats is desirable for obtaining products with higher added value and reducing negative environmental impact [25]. Fat waste includes beef tallow, which is obtained from the rendering of fatty tissue from beef cattle, mutton tallow from the rendering processing of sheep, lard from the rendering processing of pigs or chicken fat from the rendering processing of feathers, blood, skin, offal and trimmings [26,27]. 

Even though venison production represents a minority share worldwide, it is a significant share of the meat industry in many regions [28]. In the 1970s, the global trade was estimated at one million tons a year [29]. Since then, however, this production has gradually increased and now ranks around two million tons annually [30]. The fat content of venison is approximately 7–10% [31]. There are typical strong seasonal fluctuations in these animals, accumulating significant amounts of subcutaneous fat during summer and autumn. In winter, food consumption is lower, influenced by the reduced photoperiod, and the animals mobilize their body reserves [28]. Deer fat is characterized by its high content of fatty acids with a long carbon chain and its hardness [32]. Tallow is a tough fat containing little unsaturated fatty acids [33]. Processing and use of deer fat have been gaining attention [34]. Due to its stability and hardness, which is suitable for maintaining the product’s shape, it has excellent potential for developing new products in the food, pharmaceutical, and cosmetic industries [35]. 

The objectives of this study are as follows: (1) To test the possibility of hydrolysis of deer tallow by microbial lipases (in granular and liquid form) and to study the influence of selected process parameters to prepare partially hydrolyzed products; (2) To analyze the chemical, textural and optical properties of prepared samples; (3) To propose optimal conditions for enzymatic hydrolysis of deer tallow and its possible industrial applications.

## 2. Results and Discussion

Products consisting of partially hydrolyzed deer tallow containing acylglycerols and fractions of free fatty acids were prepared by microbial lipase hydrolysis. The microbial sn-1,3-specific lipases are used to hydrolyze fat in ester linkages [4].

### 2.1. Degree of Fat Hydrolysis and Chemical Analysis

The conditions of the experiment, the summary results of the chemical analyses, and the degree of hydrolysis (DH) of partially hydrolyzed deer tallow using granular (G) and liquid (L) forms of the enzyme are given in Table 1. 

The results of the analysis of variance for peroxide value, acid value, and degree of hydrolysis are given in Table 2. Contour graphs in Figure 1 and Figure 2 show the relationship between the dependent and independent variables.

The regression analysis results of the peroxide value for the fat treated with the liquid form of the enzyme did not show statistical significance of the observed factors (*p* ≤ 0.05). On the contrary, reaction time was a statistically significant factor for samples treated with the granular form of the enzyme. From the regression analysis results of the acid value for the fat hydrolyzed using the enzyme’s granular and liquid forms, it is clear that the amount of water in the reaction is a statistically significant factor. The regression analysis results of the degree of hydrolysis correlate with the results of the acid value since the degree of hydrolysis was calculated from the acid values.

The contour graph in Figure 1 shows the relationship between PV and DH and the studied variables (the amount of water, the amount of enzyme in the reaction, and the reaction time) with the G form of the enzyme. From Figure 1a, it can be seen that PV is lowest at a low amount of water in the reaction (<12.5%) and short reaction time (<2.5 h). The PV increases significantly with increasing amounts of water and time. On the contrary, Figure 1b shows that PV is lowest at a high amount of enzyme in the reaction (>5%) and short reaction time (<2.5 h). The PV increases with increasing reaction time. From Figure 1c, it can be seen that DH is highest at higher amounts of water in the reaction (>19%) and higher amounts of enzyme (>4.5%). With increasing amounts of water but the same amount of enzyme, DH decreases. The lowest DH was achieved at a lower amount of water (<10%) and a higher amount of enzyme (>4.5%). The higher esterification can explain this trend during the reaction, as the esterification reaction dominates the hydrolysis reaction at a lower water content. Figure 1d shows that the lowest DH was achieved with a lower amount of water (<10%) and a longer time (>5 h). As the amount of water in the reaction increases, the degree of hydrolysis increases. This phenomenon is attributed to the predominance of esterification reactions at longer reaction times.

From Figure 2a, it can be seen that the highest DH is achieved with a higher amount of L enzyme in the reaction (>3%) and a longer reaction time (>3 h). With a decreasing amount of water and shorter reaction time, DH decreases. Figure 2b shows that the lowest DH was achieved with lower water content (<10%) and shorter time (<4 h). With the increasing amount of water in the reaction and a longer time, DH increases. 

The lowest PV (11.91 µval/g) was for sample 10*, which was not treated with the enzyme. For samples treated with the enzyme’s G form, the PV ranged from 12.77 to 29.54 µval/g. The highest PV value was found in sample 6. For samples treated with the enzyme’s L form, the value ranged from 12.32 to 19.30 µval/g. PV should not exceed 30 µval/g in food and cosmetics products to avoid odd flavors [36]. All samples of hydrolyzed fat had PV within the generally acceptable limit. When comparing the PV results for fat treated with G and L enzymes, it was shown that treatment with L enzymes resulted in lower PV. The AV of raw deer tallow is 2.05 mg/g. The acid value for the samples treated with the granular enzyme ranged from 25.43 to 87.33 mg/g. The highest value was reached in sample 8 and the lowest in sample 4. For samples treated with the enzyme’s L form, the AV was 38.54–107.01 mg/g. The highest value was found in sample 8, and the lowest in sample 2.

Samples treated with the G enzyme had a DH in the 11.79–40.48% range, while samples treated with the L enzyme had a DH in the 17.86–49.60% range. For both sets of samples, it was found that the highest AV and DH values were obtained for sample 8. The samples treated with the L enzyme had 20% higher DH values than those treated with the G form of the enzyme. The higher DH of L lipase is otherwise due to its faster mixing in the reaction mixture, which leads to its speedier reaction at the phase interface between water and fat. For the granular form of the enzyme, its reactivity is limited due to its slower solubility and is also affected by the amount of water in the reaction. For the G enzyme, partial denaturation occurred after longer reaction times. The granular enzyme requires dry storage away from sunlight. On the contrary, a disadvantage of using L lipase is its limited durability at operating temperature (22.0 ± 3.0 °C). The price difference between the two enzyme forms is approximately 10% in favor of the L enzyme. On the other hand, the minor disadvantages of the L enzyme are compensated by the higher hydrolytic efficiency.

Ma et al. [37] tested five different enzymes in their experiments and used mutton tallow for processing. The peroxide value of the enzymatically untreated mutton tallow was 9.5 µval/g. Lipases were added at 0.55 g of enzyme/100 g of mutton tallow. The temperature was set according to the optimum enzyme temperature, and the reaction time was 3 h. After treatment, the temperature was raised to 95 °C (10 min) to deactivate the enzyme. The PV of the enzyme-treated mutton tallow ranged from 38.1–50.1 µval/g, depending on the type of lipase used. The value of the raw mutton tallow corresponds to the value found for natural deer tallow. As in the present study, an increase in PV was found after using the enzyme. The PV of enzymatically treated mutton tallow was two to three times higher than our results for enzymatically treated deer tallow. The acid value ranged from 57.40 to 77.57 mg/g. The AV of the enzymatically untreated mutton tallow was 0.68 mg/g. The AV values corresponded with those found in our experiments and are mainly influenced by the processing conditions.

In a study by Teng et al. [38], DH was measured in enzymatically hydrolyzed chicken fat. The preparation was carried out using the shake-flake method with an emulsion prepared from chicken fat and phosphate buffer in a ratio of 1:1. During the reaction, the temperature was maintained using a water bath (35–50 °C), and the amount of enzyme added was 0.2–0.71%. The degree of hydrolysis in this experiment ranged from 4.7 to 97.1%. The highest DH for this fat was almost twice as high as the highest in our study. This fact may be attributed primarily to the different enzymatic hydrolysis conditions in compared studies. From the difference in the fatty acid composition of chicken fat (total SFAs 31.5%, total MUFAs 52.9%, and total PUFAs 15.6%) [38] and deer tallow in the presented study (total SFAs 63.9%, total MUFAs 31.0%, and total PUFAs 5.1%), it can be concluded that the higher MUFAs abundance of chicken fat may account for the higher DH; highest DH 97.1% [38] versus highest DH 49.1% for L enzyme.

In a study by Carvalho et al. [39], salmon oil was enzymatically hydrolyzed by native lipases. Three different enzymes were used in the enzymatic treatment. The reaction was carried out for 6–48 h at 35–45 °C, with an enzyme addition of 100–500 U/g and different water/oil ratios. After completion of the reaction, the enzymes were deactivated at 90 °C for 15 min. Depending on the enzyme used and the reaction conditions, DH ranged from 1.0 to 57.2%. The DH increased with more water and enzyme in the reaction, confirming the significant influence of process conditions during the reaction. The DH is very similar to that in our study. Also, the trend was confirmed where DH increases with increasing amounts of water and enzymes in the reaction. From the representation of fatty acids in deer tallow in the presented study and salmon oil, no clear conclusions can be drawn about the effect of fatty acid composition and representation on DH. In the study [39], the authors report only a total for PUFAs of 30.1%, which is 5.9 times higher than the presented study. Nevertheless, the lack of complete representation of SFAs and MUFAs in [39] makes it impossible to explain the possible effect of fatty acids on the different DH values when comparing studies.

Rather than the fatty acid composition of the different raw materials, the DH and properties of the hydrolyzed fats are significantly influenced by the conditions of hydrolysis as well as the type and form of enzyme used. In studies by Teng et al. [38] and Carvalho et al. [39], microbial lipases were similarly used to hydrolyze fats. However, the physical form and activity of the enzymes used are not precisely specified in the studies, which does not allow an unambiguous comparison of the enzyme hydrolysis results; furthermore, the confrontation of the studies is complicated by the different feedstock.

### 2.2. Color

The color was measured using the International Commission on Illumination L*a*b* color space (CIEL*a*b*), where the L* indicates the specific lightness and takes values from the interval 0 (black) to 100 (white). The a* coordinate runs from green to red, and the b* coordinates from blue to yellow. In some studies [40,41], the parameter ∆E* indicates the total color difference of the samples compared to the standard (enzymatically untreated tallow). The calculated ∆E* values represent the intensity of the color change but not its direction. All the results are presented in Table 3.

The highest lightness of 90.55 was measured for sample 10* (blind sample), which was not subjected to enzymatic hydrolysis. For samples treated with the G enzyme, the lightness ranged from 86.98 to 88.96; the a* parameter ranged from −4.06 to −2.20, and the values of the b* parameter ranged from 4.88 to 16.95. For the samples that were treated with the liquid form of the enzyme, the value of parameter L ranged from 84.60 to 87.27; the a* parameter ranged from −3.48 to −2.34 for all samples, and parameter b* had the most extensive range of measured values, ranging from 4.43 to 18.44. The lowest value was measured in sample 3, and the highest was found in sample 4.

Since not enough literature sources of hydrolyzed fats were found, studies with similar matrices were used for comparison. Ye et al. [40] used beef tallow to obtain beef flavor in hydrolyzed soybean meal-based products in their research. Beef tallow was mixed in a 1:1 ratio with phosphate buffer and 0.001 g lipase/g tallow. The enzymatic hydrolysis reaction was carried out at 55 °C for eight hours with constant stirring. The temperature increased to 90 °C for 15 min to inactivate the enzyme. After the preparation of the hydrolyzed tallow, the product was slightly thermally oxidized, and subsequently, color changes occurred due to Maillard reactions. The value of ∆E* was in the range of 2.0–3.2, which almost corresponds to the results of samples treated with G enzyme. Samples treated with L enzyme had ∆E* up to four times higher than the results presented. The result corresponds well with the higher DH (up to 49.6%) of the products prepared with the L enzyme, in which the fatty acid cleavage is more significant than those prepared with the G enzyme (DH maximum 40.5%). Color was measured in a study by Ziarno et al. [41] for butter and butter substitutes. ∆E* was in the range of 12.37–16.51, where the values found correspond to the highest results for samples treated with the L form of the enzyme. Samples treated with the G form of the enzyme showed about half the value. The amount of released fatty acids can influence the color change of the studied samples. 

### 2.3. Vibrational Characterization of Functional Groups

On the FTIR records, seven characteristic peaks were found in all hydrolyzed fats; the model representation is shown in Figure 3. For comparison, five peaks were selected, with reference values and molecular action of individual peaks [42,43,44,45,46,47], which are recorded in Table 4. For these peaks, their absorbance changed the most.

When comparing the peaks, it is clear that during enzymatic hydrolysis, the relative intensity changes the most at the value of 1739 cm^−1^, corresponding to C=O stretch molecular action. The highest absorbance at this peak was found in sample 10*, which was not enzymatically treated, and the intensity was 0.387. The lowest intensity was found in sample 6, treated with L enzyme, where the absorbance was 0.192. For samples treated with the G form of the enzyme, the lowest absorbance value was for sample 8. The absorbance value at wavenumber 1739 cm^−1^ corresponds to DH, where samples with lower DH showed lower absorbance values.

A study by Salimon et al. [48] reports FTIR results for hydrolyzed *Jatropha curcas* seed oil. The reaction was carried out at temperatures varying from 50–70 °C and 1.5–2.5 h. Another factor affecting the hydrolysis was the concentration of ethanolic KOH. The measured wavenumbers corresponding to the functional groups found in this type of fat agreed with those measured in our study. Moentamaria et al. [49] published a study in which coconut oil was enzymatically hydrolyzed using immobilized lipase produced by *Mucor miehei*. The functional groups found corresponded to the wavenumbers found in the studied samples. In the study by Khaskheli et al. [50], castor oil was analyzed after enzymatic hydrolysis using lipase produced by *Rhizopus oryzae*. As in the presented study, a decrease in intensity in the region around 1746 cm^−1^ occurred after enzymatic hydrolysis due to a reduction in FA ester concentration. None of the mentioned studies looked in detail at the absorbance intensity of individual peaks, only their wavenumbers. When comparing the peaks of the purified fat and the blind sample (No. 10), it is clear that there is no significant change in the intensity of the selected peaks and no change in the peak indicating the ester bond (C=O stretch). The highest difference in intensity was found when comparing the peaks at wavenumber 1739 cm^−1^ of the purified tallow and hydrolyzed samples; in the case of sample 8 treated with the G form of the enzyme, the intensity of this peak was about half that of the purified sample.

### 2.4. Textural Properties

The following variables were defined when measuring textural properties: harness, spreadability, adhesiveness, and stickiness. All results were evaluated by regression analysis, see Table 5. No statistically significant factor was found in the regression analysis of the textural properties of the samples where the G enzyme was used; therefore, a contour graph was not generated for any of the textural properties. From the regression analysis results of samples treated with the L form of the enzyme, it can be seen that two statistically significant factors (*p* < 0.05) were found for hardness and spreadability, namely the amount of water in the reaction and the reaction time. For stickiness, only one statistically significant factor, namely the amount of water in the reaction, was found. Regression analysis of adhesiveness shows that no statistically significant factor was found. 

Figure 4 shows the relationship between the response variables (hardness, spreadability, adhesiveness, and stickiness) and the two predictive variables (amount of water and reaction time). In Figure 4a, it can be seen that the lowest hardness was achieved with lower water content (<15%) and short reaction time (<4 h). The hardness also increases as the reaction time and the amount of water increases. In Figure 4b, it can be seen that spreadability behaves similarly to hardness, with the lowest values reached at lower amounts of water in the reaction (<12.5%) and shorter reaction time (<3.5 h). The spreadability increases as the amount of water and the reaction time increase. In Figure 4c, adhesiveness concerning the amount of water and reaction time is shown using a contour graph. It can be seen that at lower water amounts (<18.5%) and shorter time (<3.5 h), the adhesiveness is highest and decreases with increasing water amount and longer reaction time. When stickiness was measured, with the results shown in contour Figure 4d, it was found that it behaves similarly to adhesiveness, reaching the highest values with lower water content < 17.5% and shorter reaction time (<3.5 h). Whereas, as the water content and the reaction time increase, this property decreases.

Textural properties are most often measured on food matrices, such as butter. Textural properties provide essential information on the properties and composition of products. Texture is thus an important indicator for many customers as it influences the use of products. In a study by Ziarno et al. [41], the textural properties of butter and butter substitutes were observed. The textural properties were measured at varying temperatures. For comparison, a temperature of 4 °C was chosen where the spreadability ranged from 13.21 N/s to 94.62 N/s, similar to the values found in the studied samples. The hardness ranged from 2.61 to 19.28 N, and our value was up to twice as high as the highest value in the study [41]. The determined stickiness was in the range of 1.01–5.03 N, with all the tested samples’ values corresponding to our study’s values. The adhesiveness ranged from 3.92 to 18.46 N/s, up to 100 times lower, indicating a much poorer spreadability of the studied samples. The different results may be due to the various types of samples tested. 

### 2.5. Proposal of Optimal Conditions for Enzymatic Hydrolysis of Deer Tallow and Contribution of the Study to the Praxis 

The correct setting of the enzymatic hydrolysis conditions is vital in terms of the degree of hydrolysis and the properties of the prepared products. By the amount of water in the reaction, the amount of enzyme, and the reaction time, up to 40% DH (for the G enzyme) or 50% (for the L enzyme) of the tallow can be achieved. The oxidative stability of the modified products is high. The PV values meet the limits for food and cosmetic applications; the lowest PV values were obtained using the L enzyme. In the case of cosmetic applications of hydrolyzed tallow, the addition of 24% (*w*/*w*) water to the reaction (based on the fat amount), the addition of 6% (*w*/*w*) L enzyme, and a reaction time of 2 h can be recommended as optimal conditions. Under these conditions, the prepared product (DH 44%) exhibits a coloring that does not require the addition of optical brighteners to manufacture the cosmetic product. The shorter reaction time is also preferable regarding textural properties, where these samples have better spreadability, which may contribute to a more comfortable application from the consumer’s point of view. Hydrolyzed deer tallow is a preferred ingredient in traditional consumer formulations of cosmetic products such as balms, ointments, or creams due to its proven effects on healing the skin or addressing skin conditions. Another aspect of their application may be cultural, religious, or other consumer preferences, but always concerning quality, purity, safety, and health of the end consumer without negatively affecting the environment and without losing biodiversity, integrity, and sustainability of the ecosystem functions [51,52,53]. In the case of pharmaceutical applications of hydrolyzed tallow, the addition of 24% (*w*/*w*) water to the reaction (based on the fat amount), the addition of 6% (*w*/*w*) liquid enzyme, and a reaction time of 6 h can be recommended as optimal conditions. Under these conditions, the prepared product (DH 50%) has a more remarkable color change, but this is not a significant deficiency. Compared to chemicals commonly used in the chemical hydrolysis of fats, e.g., CH_3_ONa using Na-K alloy or sodium alcoholate catalysts [54,55], microbial lipase costs about 5 times more. However, the higher cost of the enzyme is compensated by the lower temperature during hydrolysis and the ease of inactivation by increasing the temperature of the reaction mixture. In chemical hydrolysis, the inactivation of a reaction is conducted by adding water to the reaction mixture, which would be unwanted for the intended use of our prepared products. The use of enzymatic hydrolysis also has a lower environmental impact.

## 3. Materials and Methods

### 3.1. Materials, Equipment and Chemicals

The tallow from the European fallow deer (*Dama dama*) was supplied by Venison CZ Ltd. (Míškovice, Czech Republic). First, conventional food processing methods were used to analyze the raw material [56,57,58,59,60,61,62]. Dry matter content is 90.2 ± 1.3%; proteins in the dry matter are 3.1 ± 0.2%, and inorganic matter is 0.14 ± 0.06%. After the purification of the fat, the primary analyses were performed will the following findings: acid value 2.6 ± 0.2 mg/g; saponification value 218.3 ± 1.2 mg/g; peroxide value 27.81.1 µval/g of fat; and iodine value 12.5 ± 0.4 g/100 g. Each analysis was determined three times; arithmetic mean values and standard deviations were calculated using Microsoft Office Excel 2016 (Denver, CO, USA). The fatty acid composition of deer fat is listed in Table 6. Saturated fatty acids accounted for 63.9% of the total fatty acids; the percentage of unsaturated fatty acids was about 36.1%, in which monounsaturated and polyunsaturated fatty acids were 31.0 and 5.1%, respectively. The qualitative composition of fatty acids corresponds to the literature data of the same type of raw material [63,64]. Minor variations in the quantitative composition can be caused by several factors, such as the method of rearing (conventional or organic farms), differences in nutrition, feeding methods, or the animal’s sex.

The following equipment was used: electronic scales Kern 440-47 (KERN & SOHN GmbH, Balingen, Germany), a Braher P22/82 meat mincer (Braher, San Sebastian, Spain), heating plate (Schott Geräte, Mainz, Germany), an UltraScan VIS Spectrophotometer HunterLab (HunterLab, Reston, VA, USA), a TA.XTplus Texture Analyzer (Stable Micro Systems Ltd., Godalming, UK), a compact FT-IR spectrometer ALPHA II BRUKER,(BRUKER, Billerica, MA, USA), and a Shimadzu Model GC-14A Gas Chromatograph (Shimadzu Europa GmbH, Duisburg, Germany). Lipex^®^ Evity 200 L (Novozymes, Copenhagen, Denmark) is a liquid enzyme produced by the fermentation of microorganisms. Lipex ^®^ Evity 100 T (Novozyme, Copenhagen, Denmark) is a granular enzyme produced by the fermentation of microorganisms. Enzymes are not GMO.

### 3.2. Experimental Design and Statistical Analysis

The design of the experiment (DOE) allows us to examine how process factors (independent variables) affect the dependent variables. Using DOE, it is possible to identify statistically significant process factors and perform process optimization [65]. Various planned experiments are commonly used, including a three-level complete factorial design, a central composite design, a Box–Behnken design, or a Taguchi design [66]. Based on our preliminary results and the results of studying processing conditions of similar studies [37,38,39,40], the critical reaction factors are the amount of water and enzyme in the reaction and the reaction time. Our study applied the Taguchi design with three independent variables studied at minimum and maximum levels with one central experiment. The following were chosen as independent variables, with factor levels: factor A-amount of water (8, 16, 24%); factor B-amount of the enzyme (2, 4, 6%); factor C-reaction time (2, 4, 6 h). The dependent variables observed were PV, AV, DH, color, and textural properties. Furthermore, the functional groups were analyzed. 

Fat analysis was performed in four repetitions, and mean values and standard deviations were calculated using Microsoft Office Excel 2016 (Microsoft, Denver, CO, USA). Regression analysis of the data was performed using the statistical software Minitab^®^ 18.1 (Fujitsu Ltd., Tokyo, Japan). Analysis of variance (ANOVA) was determined for PV and DH. Statistical significance of the factors was assessed at the significance level of 95% (*p* ≤ 0.05); factors with a value ≤ 0.05 affect the process variables. Graphical analysis of the data was performed using the same software to create contour graphs showing the relationships between the dependent and independent variables by applying Akima’s polynomial method of interpolation.

### 3.3. Processing of Deer Tallow

The block diagram shows a scheme for processing deer tallow into a hydrolyzed product (see Figure 5).

Homogenization and purification. The obtained fat tissue contained a significant amount of muscle proteins, and therefore, homogenization and purification were necessary. The raw material was ground using a meat cutter (power 300 kg/h) through a cutting system of perforated plates with kidney-shaped holes (20 mm diameter) and then through a plate with 13 mm holes. A double-sided knife was used between the cutting plates. The ground raw material was melted at 70.0 ± 1.0 °C in an oven for 2 h. After melting, the fat was filtered hot through several layers of cloth to obtain pure fat. The separated proteins and other impurities accounted for about 30% by weight.

Hydrolysis of fat was conducted using a batch process. Water in the amount according to factor A (8; 16; 24%) and enzyme in the amount according to factor B (2; 4; 6%) were added into the beaker of a 150 mm diameter and mixed at 450 ± 100 rpm at 23.0 ± 0.5 °C for 3 min. Further, the pure fat was added, and the mixture was placed in a water bath at a temperature of 50.0 ± 0.5 °C and stirred at 450 ± 100 rpm for a time according to the factor C (2; 4; 6 h). After hydrolysis, the enzyme was inactivated by heating the mixture at a rate of dt/dτ 6 °C/min to 85.0 ± 0.5 °C and held at this temperature for 5 min. Subsequently, silica gel (in the amount of 20%, based on fat weight) was added, and the mixture was stirred at 450 ± 100 rpm for 3 min and filtered through Filpap KA-2 filter paper (Filpap, Ixelles, Belgium) on a Büchner funnel; before filtration step, the laboratory glassware was heated to about 100 °C. Subsequently, the fat was left in a beaker at a laboratory temperature of 21.0 ± 0.5 °C for 1 h and then placed in an incubator (4.0 ± 0.2 °C). Characterization of properties (PV, AV, color, functional groups, and textural properties) DH was calculated from the measured values.

### 3.4. Analytical Part

Dry matter was determined by gravimetric method [56]. The standard Kjeldahl method was used to determine nitrogen content [57]. The inorganic content was determined gravimetrically after incineration of the sample [58]. Furthermore, AV, SV, and iodine values were determined according to the test methods for oils and fats. Since these are commonly used techniques, only the principles are provided. The peroxide value is given in milliequivalents of peroxide per kg of oil capable of releasing iodine from potassium iodide during the test; iodine is further estimated using a standard sodium thiosulphate solution [59]. The acid value is an indicator of free FA content in the fat. It is expressed as the mass of potassium hydroxide in mg required to neutralize free FAs in one gram of fat. It is independent of molecular weight, almost twice the proportion of free FAs when expressed on an oleic acid basis [60]. The saponification value is an indicator of the content of all FAs. It is described in mg of potassium hydroxide, which is needed to saponify 1 g of fat. This value includes the neutral fat and free FAs present and relates to the molecular weights of the respective FAs [61]. The iodine value (IV) measures lipid unsaturation and is given as the amount of halogen, most commonly iodine, in grams that can be added per 100 g of fat [62]. The fatty acid composition was determined by gas chromatography according to the standard methodology used to analyze animal and vegetable fats and oils [67]. 

The hydrolysis degree of *DH* (%) was determined according to the American Oil Chemists’ Society methods [68]. The determination was carried out by measuring the acid value *AV* of original fat (*O*) and hydrolyzed fat (*H*), and the values were calculated according to Equation (1):(1)$$DH\;\left(\%\right)=\frac{{AV}_{H}}{{SV}_{O}-{AV}_{O}}\times 100$$

The color measurement was performed using the methodology of [69] using the HunterLab UltraScan VIS Spectrophotometer. Calibration was conducted on a black-and-white background. Subsequently, measurements of individual samples were taken. From the measured values, the total color difference (∆*E**) was calculated using Formula (2):(2)$${∆E}^{*}=\sqrt{{∆L}^{*2}+{∆a}^{*2}+{∆b}^{*2}}$$
where ∆*L** is the component difference of *L** between the unhydrolyzed tallow and hydrolyzed sample; ∆*a** is the component difference of *a** between the unhydrolyzed tallow and hydrolyzed sample; ∆*b** is the component difference of *b** between the unhydrolyzed tallow and hydrolyzed sample.

Vibrational characterization of functional groups in the tested samples was performed according to Official Methods of Analysis [70] using an ALPHA II compact FTIR spectrometer with a range of 400–4000 cm^−1^ with a resolution of 8 cm^−1^ using 32 scans. All spectra were corrected against the air spectrum. After each scan, new measurements of reference air and background were taken. The spectra were recorded as absorbance values. The graphical presentation of the results is expressed by the dependence of absorbance (au—arbitrary units) on wavenumbers (cm^−1^).

The textural properties were measured by a modified method applied to semi-rigid matrices [71] using the TA.XTplus Texture Analyzer (Stable Micro Systems Ltd., Godalming, UK). The measurement was made penetrometrically using the HDP/SR probe at a depth of 2 mm for a given sample at a speed of 1 mm/s. Samples were melted and poured into dishes 24 h before measurement and left at 21.0 ± 0.5 °C for two h. Subsequently, the dishes with the samples were stored at 4.0 ± 0.2 °C for 22 h and cleaned before measurement. Each sample was measured four times. The average values of the textural parameters (hardness, spreadability, stickiness, and adhesiveness) and standard deviations were determined from the measured values.

## 4. Conclusions

Deer tallow has almost no use nowadays, even though it contains valuable fatty acids. The advantage of its processing lies in the relatively low melting point, which is vital in terms of energy savings and better quality of the processed fat. Enzymatic treatment can yield partially hydrolyzed tallow, which can be used to produce various products. The amount of water in the reaction affects the DH, and the process conditions of the reaction affect the textural properties and color of the prepared samples. Changes at the molecular level occur during the reaction, manifested by a change in the intensity of the ester bond peak. A low PV confirmed the oxidative stability of the prepared products. Under optimal reaction conditions, enzymatically modified fats suitable for cosmetic, pharmaceutical, and food applications can be prepared.

## Figures and Tables

**Figure 1 ijms-25-04002-f001:**
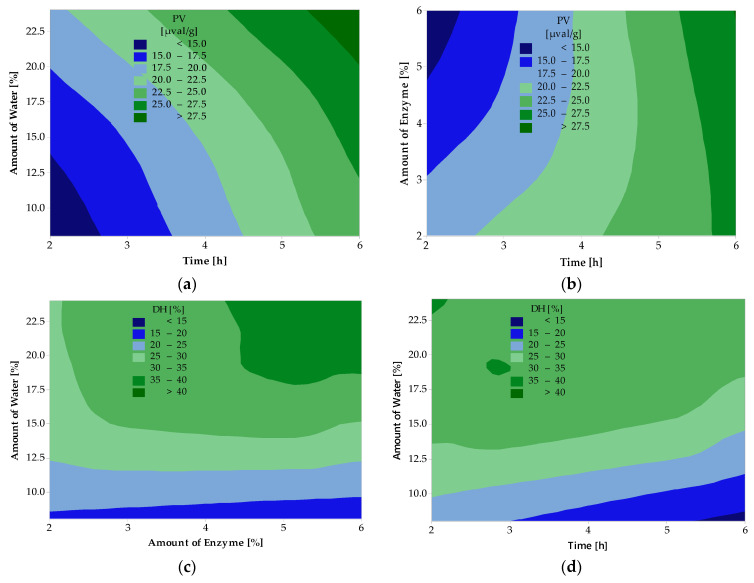
Effects of water amount and reaction time (**a**) and enzyme amount and reaction time (**b**) on PV of hydrolyzed deer tallow; effects of water amount and enzyme amount (**c**) and water amount and reaction time (**d**) on DH of hydrolyzed deer tallow for samples treated with the G form of the enzyme.

**Figure 2 ijms-25-04002-f002:**
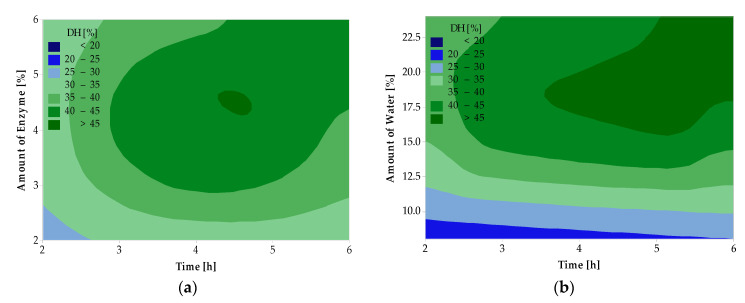
Effect of enzyme amount and reaction time (**a**) and water amount and reaction time (**b**) on DH of hydrolyzed deer tallow treated with the L form of the enzyme.

**Figure 3 ijms-25-04002-f003:**
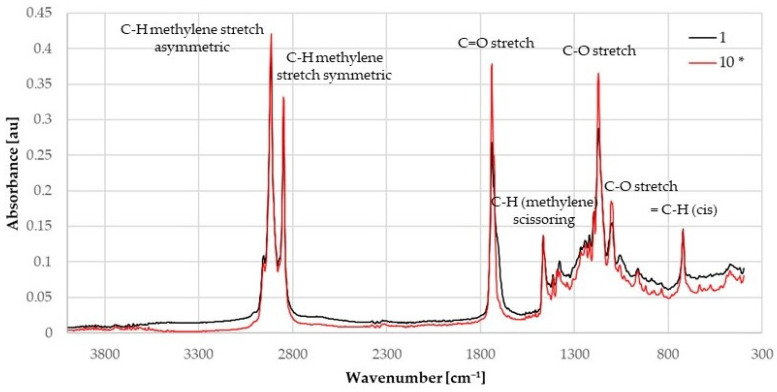
The FTIR spectra of sample 1 (G form of the enzyme), and 10 (blind sample).

**Figure 4 ijms-25-04002-f004:**
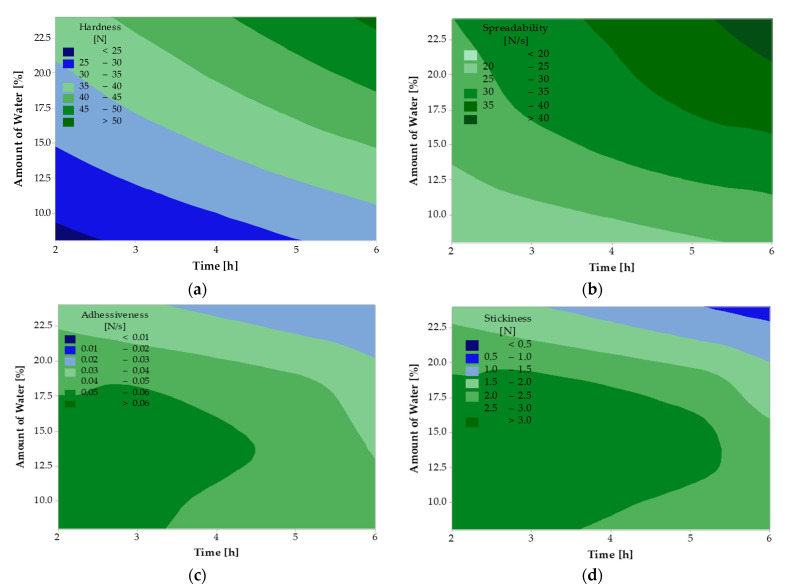
Effect of the amount of water in the reaction and time on the textural properties of hydrolyzed deer tallow when using the L form of the enzyme; (**a**)—hardness, (**b**)—spreadability, (**c**)—adhesiveness, (**d**)—stickiness.

**Figure 5 ijms-25-04002-f005:**
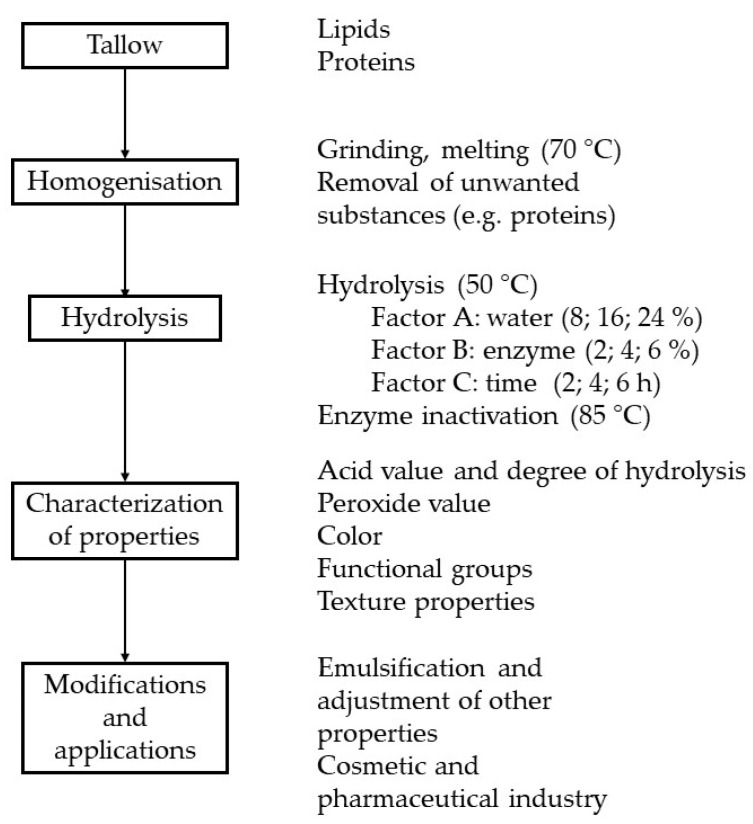
Flow chart of preparation of a hydrolyzed product from deer tallow.

**Table 1 ijms-25-04002-t001:** Schedule of experiments and results of chemical analyses and degrees of hydrolysis of hydrolyzed deer tallow by G and L forms of the enzyme.

Exp. No.	Factor A [%]	Factor B [%]	Factor C [h]	Peroxide Value [µval/g]		Acid Value [mg/g]		Degree of Hydrolysis [%]	
				G	L	G	L	G	L
1	8	2	2	13.65 ± 0.25	16.20 ± 0.41	49.30 ± 1.60	51.64 ± 0.95	22.85	23.93
2	8	2	6	21.66 ± 1.20	19.30 ± 0.61	33.14 ± 0.03	38.54 ± 0.50	15.36	17.86
3	8	6	2	12.77 ± 0.28	17.36 ± 0.42	46.17 ± 1.90	40.42 ± 0.82	21.40	18.73
4	8	6	6	26.46 ± 0.96	14.88 ± 1.70	25.43 ± 0.52	68.98 ± 0.74	11.79	31.97
5	24	2	2	24.41 ± 0.72	12.32 ± 0.40	69.82 ± 1.90	67.86 ± 1.80	32.36	31.46
6	24	2	6	29.54 ± 0.74	15.84 ± 0.90	57.81 ± 1.00	96.55 ± 1.10	26.80	44.75
7	24	6	2	14.49 ± 0.23	17.24 ± 0.80	81.48 ± 1.40	94.02 ± 0.54	37.77	43.58
8	24	6	6	28.75 ± 1.00	15.37 ± 0.82	87.33 ± 0.37	107.01 ± 0.18	40.48	49.60
9	16	4	4	20.58 ± 0.15	14.13 ± 0.59	69.28 ± 1.00	94.93 ± 1.40	32.11	44.00
10 *	16	0	4	11.91 ± 1.90		2.03 ± 0.42		-	

Factor A—amount of water in the reaction, Factor B—amount of enzyme in the reaction, Factor C—reaction time, G—granular form of the enzyme, L—liquid form of the enzyme, *—blind experiment (without enzyme treatment).

**Table 2 ijms-25-04002-t002:** ANOVA table for results PV, AV, and DH of deer tallow using G and L form of the enzyme.

	Degree of Freedom	Sum of Squares		Mean Squares		*p*-Value	
Type of Enzyme		G	L	G	L	G	L
Regression equations	G: PV = 7.13 + 0.354 A − 0.424 B + 2.568 C						
	L: PV = 16.73 − 0.109 A + 0.074 B + 0.142 C						
Regression	3	280.939	6.8937	93.646	2.2979	0.022	0.733
Factor A: Water [%]	1	64.128	6.0726	64.128	6.0726	0.064	0.329
Factor B: Enzyme [%]	1	5.763	0.1770	5.763	0.1770	0.509	0.861
Factor C: Time [h]	1	211.049	0.6441	211.049	0.6441	0.008 *	0.739
Error	5	57.025	25.9342	11.405	5.1868		
Total	8	337.964	32.8279				
Regression equations	G: AV = 25.3 + 2.225 A + 1.90 B − 2.69 C						
	L: AV = 3.6 + 2.592 A + 3.49 B + 3.57 C						
Regression	3	2881.6	4236.6	960.5	1412.2	0.031	0.039
Factor A: Water [%]	1	2534.7	3438.7	2534.7	3438.7	0.008 *	0.012 *
Factor B: Enzyme [%]	1	115.1	389.8	115.1	389.8	0.403	0.249
Factor C: Time [h]	1	231.8	408.1	231.8	408.1	0.252	0.239
Error	5	690.9	1144.3	138.2	228.9		
Total	8	3572.5	5380.9				
Regression equations	G: DH = 11.74 + 1.031 A + 0.879 B − 1.247 C						
	L: DH = 16.73 − 0.109 A + 0.074 B + 0.142 C						
Regression	3	619.16	910.57	206.39	303.52	0.031	0.039
Factor A: Water [%]	1	544.67	739.20	544.67	739.20	0.008 *	0.012 *
Factor B: Enzyme [%]	1	24.75	83.72	24.75	83.72	0.403	0.249
Factor C: Time [h]	1	49.75	87.65	49.75	87.65	0.252	0.239
Error	5	148.36	245.78	29.67	49.16		
Total	8	767.52	1156.35				

*—statistically significant factor (*p*-value ≤ 0.05), G—granular form of the enzyme, L—liquid form of the enzyme, PV—peroxide value, AV—acid value, DH—degree of hydrolysis.

**Table 3 ijms-25-04002-t003:** Values of color parameters for hydrolyzed deer tallow.

Exp. No.	L*		a*		b*		∆E*	
	G	L	G	L	G	L	G	L
1	88.32 ± 0.31	84.91 ± 1.08	−2.71 ± 0.14	−3.32 ± 0.11	5.38 ± 0.16	17.06 ± 0.52	2.43	12.20
2	87.75 ± 0.30	86.19 ± 0.83	−2.20 ± 0.23	−2.66 ± 0.22	4.88 ± 0.24	5.74 ± 0.50	3.14	4.40
3	87.46 ± 0.48	87.26 ± 0.27	−3.46 ± 0.23	−2.97 ± 0.06	9.17 ± 0.69	5.43 ± 0.35	4.38	3.46
4	88.80 ± 0.48	84.60 ± 0.31	−2.98 ± 0.06	−2.81 ± 0.19	6.80 ± 0.45	18.44 ± 1.16	1.94	13.54
5	88.01 ± 0.35	86.06 ± 0.05	−2.29 ± 0.57	−3.06 ± 0.01	6.51 ± 0.78	14.76 ± 0.06	4.09	9.62
6	87.74 ± 0.56	85.90 ± 0.41	−3.29 ± 0.18	−3.41 ± 0.07	9.09 ± 0.68	13.38 ± 0.85	4.09	8.55
7	87.04 ± 0.30	86.36 ± 0.66	−3.45 ± 0.20	−2.40 ± 0.10	16.95 ± 0.53	17.39 ± 0.14	11.29	11.87
8	88.96 ± 0.30	86.28 ± 0.50	−4.06 ± 0.12	−3.15 ± 0.58	13.76 ± 1.08	13.35 ± 0.88	7.83	8.30
9	86.98 ± 0.06	87.27 ± 0.56	−3.40 ± 0.15	−3.48 ± 0.31	11.12 ± 0.77	10.60 ± 0.52	6.11	5.54
10 *	90.55 ± 0.21		−2.34 ± 0.08		6.28 ± 0.41		-	

L*, a*, b*—color parameters, ∆E*—total color difference of samples, G—granular form of the enzyme, L—liquid form of the enzyme, *—blind experiment (without enzyme treatment).

**Table 4 ijms-25-04002-t004:** Values of significant peaks and their reference values and molecular action of hydrolyzed deer tallow and purified deer tallow.

Wavenumber [cm^−1^]	721		1173		1739		2850		2916	
Reference Value [cm^−1^]	700–900		1000–1200		1700–1750		2830–2850		2900–2950	
Molecular Action	=C-H (cis)		C-O Stretch		C=O Stretch		C-H Methylene Symmetric		C-H Methylene Asymmetric	
Type of Enzyme	G	L	G	L	G	L	G	L	G	L
1	0.146	0.146	0.288	0.273	0.268	0.250	0.304	0.305	0.389	0.392
2	0.147	0.147	0.297	0.286	0.275	0.261	0.308	0.304	0.392	0.388
3	0.153	0.145	0.293	0.292	0.269	0.267	0.327	0.310	0.421	0.395
4	0.149	0.150	0.327	0.252	0.312	0.219	0.323	0.293	0.412	0.375
5	0.157	0.152	0.254	0.234	0.223	0.201	0.295	0.314	0.379	0.404
6	0.148	0.154	0.250	0.225	0.221	0.192	0.300	0.307	0.384	0.394
7	0.151	0.148	0.250	0.209	0.212	0.197	0.315	0.315	0.404	0.400
8	0.148	0.153	0.231	0.217	0.206	0.209	0.302	0.320	0.387	0.408
9	0.146	0.145	0.264	0.221	0.234	0.189	0.301	0.310	0.386	0.399
10 *	0.149		0.376		0.387		0.342		0.434	
Deer tallow	0.151		0.376		0.390		0.350		0.446	

G—granular form of the enzyme, L—liquid form of the enzyme, *—blind experiment (without enzymatic treatment).

**Table 5 ijms-25-04002-t005:** ANOVA table for the results of textural properties of hydrolyzed deer tallow.

	Degree of Freedom	Sum of Squares		Mean Squares		*p*-Value	
Type of Enzyme		G	L	G	L	G	L
Regression equations	G: Hardness = 25.98 − 0.179 A + 1.471 B + 0.574 C						
	L: Hardness = 7.30 + 1.0303 A + 0.404 B + 2.676 C						
Regression	3	96.15	777.921	32.05	259.307	0.287	0.000
Factor A: Water [%]	1	16.39	543.510	16.39	543.510	0.398	0.000 *
Factor B: Enzyme [%]	1	69.21	5.216	69.21	5.216	0.116	0.283
Factor C: Time [h]	1	10.56	229.194	10.56	229.194	0.492	0.001 *
Error	5	96.04	18.052	19.21	3.610		
Total	8	192.19	795.973				
Regression equations	G: Spreadability = 20.29 − 0.112 A + 1.168 B + 0.651 C						
	L: Spreadability = 6.22 + 0.812 A + 0.432 B + 2.220 C						
Regression	3	63.634	501.435	21.211	167.145	0.216	0.002
Factor A: Water [%]	1	6.444	337.740	6.444	337.740	0.459	0.001 *
Factor B: Enzyme [%]	1	43.618	5.986	43.618	5.986	0.091	0.392
Factor C: Time [h]	1	13.572	157.709	13.572	157.709	0.297	0.005 *
Error	5	50.013	34.066	10.003	6.813		
Total	8	113.647	535.500				
Regression equations	G: Stickiness = 2.59 + 0.0117 A − 0.083 B − 0.109 C						
	L: Stickiness = 3.250 − 0.0723 A + 0.154 B − 0.187 C						
Regression	3	0.67424	4.5597	0.22475	1.5199	0.928	0.104
Factor A: Water [%]	1	0.07031	2.6796	0.07031	2.6796	0.839	0.055
Factor B: Enzyme [%]	1	0.22111	0.7626	0.22111	0.7626	0.720	0.241
Factor C: Time [h]	1	0.38281	1.1175	0.38281	1.1175	0.639	0.168
Error	5	7.67336	2.1556	1.53467	0.4311		
Total	8	8.34760	6.7154				
Regression equations	G: Adhesiveness = 0.0419 + 0.000156 A − 0.00062 B − 0.00188 C						
	L: Adhesiveness = 0.0639 − 0.001250 A + 0.00250 B − 0.00375 C						
Regression	3	1.38 × 10^−4^	1.450 × 10^−3^	4.6 × 10^−5^	4.83 × 10^−4^	0.923	0.048
Factor A: Water [%]	1	1.3 × 10^−5^	8.00 × 10^−4^	1.3 × 10^−5^	8.00 × 10^−4^	0.846	0.029 *
Factor B: Enzyme [%]	1	1.2 × 10^−5^	2.00 × 10^−4^	1.2 × 10^−5^	2.00 × 10^−4^	0.846	0.192
Factor C: Time [h]	1	1.13 × 10^−4^	4.50 × 10^−4^	1.13 × 10^−4^	4.50×10^−4^	0.565	0.073
Error	5	1.485 × 10^−3^	4.39 × 10^−4^	2.97 × 10^−4^	8.8 × 10^−5^		
Total	8	1.622 × 10^−3^	1.889 × 10^−3^				

*—statistically significant factor, G—granular form of the enzyme, L—liquid form of the enzyme.

**Table 6 ijms-25-04002-t006:** Fatty acid composition for deer tallow.

Fatty Acid	Amount [%]	Fatty Acid	Amount [%]	Fatty Acid	Amount [%]
C 13:0	0.8 ± 0.06	C 16:0	31.8 ± 0.12	C 18:0	20.3 ± 0.16
C 14:0	5.4 ± 0.09	C 16:1	2.2 ± 0.04	C 18:1	18.6 ± 0.15
C 14:1	0.5 ± 0.15	C 16:3	0.5 ± 0.02	C 18:2	1.4 ± 0.04
C 14:2	0.3 ± 0.04	C 16:4	1.2 ± 0.07	C 20:0	0.9 ± 0.09
C 14:3	0.5 ± 0.07	C 17:0	2.7 ± 0.13	Σ SFAs	63.9
C 15:0	2.0 ± 0.06	C 17:1	0.6 ± 0.11	Σ MUFAs	31.0
C 15:1	9.1 ± 0.14	C 17:3	1.2 ± 0.03	Σ PUFAs	5.1

SFAs—saturated fatty acids, MUFAs—monounsaturated fatty acids, PUFAs—polyunsaturated fatty acids.

## Data Availability

Data is contained within the article.

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
