# Peer review of "Study of Processing Conditions during Enzymatic Hydrolysis of Deer By-Product Tallow for Targeted Changes at the Molecular Level and Properties of Modified Fats"

_ijms, 2024, doi:10.3390/ijms25074002_

Round 1

Reviewer 1 Report

Comments and Suggestions for Authors

The article is very well prepared in terms of content. The logical layout of the article, research methods described in great detail, which will allow the research methodology to be reproduced by other authors. The material contained is innovative due to environmental protection aspects, low energy consumption, and rational management of waste generated in the meat industry. The authors' method of using deer tallow, an almost useless meat, is a specific solution and indicates its application in the cosmetics and pharmaceutical industries. Very interesting work, it may be of interest to other researchers and readers. No substantive errors were found.

Author Response

Dear Sir / Madam,

Thank you very much for your positive comments regarding our manuscript.

Yours faithfully,

Pavel Mokrejš (corresponding author)

Reviewer 2 Report

Comments and Suggestions for Authors

In this peer-reviewed article, the authors focus on the study of processing conditions during the enzymatic hydrolysis of deer by-product tallow for targeted changes at the molecular level and in the properties of modified fats. The objectives of the study (to test the possibility of hydrolysing deer tallow using microbial lipases in order to study the influence of certain process parameters, and to analyse the chemical, textural and optical properties of the samples in order to propose optimal conditions for possible industrial applications) are perfectly presented by means of an introduction that focuses on the essentials and avoids redundant aspects.

- The study made it possible to transform tallow into hydrolyzed products; but what are these products? Certainly a Taguchi made it possible to optimize the conversion of tallow into hydrolyzed products; but their nature must be defined. The authors show changes at the molecular level observed by FTIR with a decrease in ester bonds; At this stage, scientific facts identifying the compounds obtained are lacking. Has an initial chemical (lipid) analysis of the tallow been carried out?

- The authors must specify the molecular changes by characterizing the molecules obtained after hydrolysis.

- When the authors say in abstract "Enzymatically hydrolyzed deer tallow is suitable for the preparation of cosmetics and pharmaceutical matrices". Admittedly, the compounds are of natural origin, but derived from animal biomass; this poses no problem for cosmetic applications, which are currently more industries oriented towards plant or algal biomass.

- No comments on iThenticate report

Author Response

Dear Sir / Madam,

Thank you very much for your comments regarding our manuscript. We did our best to revise our paper according to your comments, suggestions, and comments and suggestions from another reviewer. Changes in the manuscript are made in red color.

Editing of the English language was done as well.

Nine new references were added to the revised manuscript; they are made in red.

Point-by-point responses to your questions and suggestions are included below.

Yours faithfully,

Pavel Mokrejš (corresponding author)

--------------------------------------------------

Comment and suggestion 1: The study made it possible to transform tallow into hydrolyzed products; but what are these products? Certainly a Taguchi made it possible to optimize the conversion of tallow into hydrolyzed products; but their nature must be defined. The authors show changes at the molecular level observed by FTIR with a decrease in ester bonds; At this stage, scientific facts identifying the compounds obtained are lacking. Has an initial chemical (lipid) analysis of the tallow been carried out?

Response: The hydrolyzed products chemically combine residual acylglycerols and released fatty acid fractions. This fact is mentioned in the revised manuscript in the first sentence of the chapter "2. Results and Discussion". In this sense, we have also slightly modified the specific objectives of our study to make it clear to the reader what products are the target of the preparation (see page 2 of the manuscript, last paragraph). Because we did not focus on separating products from fat hydrolysis, the fatty acid composition of these reaction products was not analyzed. However, identifying fatty acids in the raw material and purified deer tallow was carried out. The results of determining fatty acids are given in the chapter "3.1. Materials, Equipment and Chemicals" in Table 6. The procedure for determining fatty acids is then newly given in chapter "3.4. Analytical Part". In addition, a further chemical analysis of the feedstock has been added. Newly, the differences in the fatty acid composition of our deer tallow and other fats (beef, chicken, salmon) in the compared studies (Refs. 38, 39, 40) and the influence of the enzymes used on the results of the hydrolysis process (degree of hydrolysis and product properties) are discussed in chapters "2.1. Degree of Fat Hydrolysis and Chemical Analysis" and "2.2. Color“.

Comment and suggestion 2: The authors must specify the molecular changes by characterizing the molecules obtained after hydrolysis.

Response: The study's objectives focused on enzymatic hydrolysis of deer tallow, which was to prepare partially hydrolyzed products containing acylglycerols and fractions of free fatty acids. The separation of these products and the analysis of the fatty acid composition of the two cleaved fractions were not carried out, as the further application of the prepared hydrolyzed products does not foresee their separation. Instead, we focused on the qualitative and quantitative study of the characteristic functional groups formed after hydrolysis. This was an essential characteristic at the molecular level corresponding to the hydrolysis degree. It is directly related to the textural properties of the hydrolyzed products, which are crucial for their subsequent processing for cosmetic and pharmaceutical products of desired rheological properties. In this sense, the results of FTIR peaks of purified fat have been complemented - see Table 4 and extended the discussion of the results in Chapter „2.3. Vibrational Characterization of Functional Groups“.

Comment 3: When the authors say in abstract "Enzymatically hydrolyzed deer tallow is suitable for the preparation of cosmetics and pharmaceutical matrices". Admittedly, the compounds are of natural origin, but derived from animal biomass; this poses no problem for cosmetic applications, which are currently more industries oriented towards plant or algal biomass.

Response: We agree with the comment that most cosmetic matrices are currently based on using plant materials and algae. However, there are still producers in the global cosmetic market whose portfolio includes cosmetic formulations based on animal raw materials with proven effects on skin healing or in addressing problematic skin conditions. These are ingredients derived from insects, terrestrial animals, or aquatic animals. Cosmetic products perform several functions - they soften and moisturize the skin, repair the skin barrier, improve elasticity, slow down the formation of wrinkles, etc. From insects, mainly bee products (honey, royal jelly, pollen, wax, etc.) are widely used; other interesting substances obtained from insects are chitin and sericin. Animal sources include, e.g. snail slime, keratin, camel milk, lanolin, and tallow. Chitin, or collagen, is also obtained from aquatic animals. This is partly due to some traditional, historically used formulations for specific cosmetic product segments (e.g., lipsticks) but also due to cultural, religious, or other consumer preferences. This is also reiterated in the chapter „2.5. Proposal of Optimal Conditions for Enzymatic Hydrolysis of Deer Tallow and Contribution of the Study to the Praxis“.

Reviewer 3 Report

Comments and Suggestions for Authors

This manuscript describes the enzymatic hydrolysis of deer by-product tallow using granular and liquid lipase. The influence of water, enzyme concentration and reaction time has been investigated, using a Taguchi design (3 factors, 3 levels). Degree of hydrolysis, oxidative stability, free fatty acids, color and textural properties of the obtained products were assessed.

The article is clearly written and the obtained results could be of interest. The methods are generally well explained, and discussion of the results is sound and supported by suitable references. Statistical analysis of the results is carried out, and the figures and tables are well presented and help to understand the work.

In summary, the article could be suitable for publication in this journal, after major revision considering the following aspects:

- The differences between using granular and liquid lipase are not sufficiently discussed. Please expand on this aspect, compare the results and highlight advantages and disadvantages of each.

- Comment on the cost of the enzymatic hydrolysis, compared to other methods.

- Explain why this specific range of the studied variables was chosen.

- When comparing the results obtained with deer tallow with those reported for other sources (e.g. beef, poultry, salmon), discuss if the composition differences in the raw materials could explain the results. Also, expand on the different enzymes used in each case.

- Write caption of Figure 3 in a more impersonal style (e.g. "FTIR spectra of samples 1 and *10, etc." instead of "The FTIR spectra of sample 1 were treated, etc."). The provided information is adequate, but the structure of the phrase should be modified.

- The methods for processing the deer tallow should be explained in more detail.

- A list of abbreviations would be useful.

Comments on the Quality of English Language

Revise the manuscript for spelling and grammar mistakes.

Author Response

Dear Sir / Madam,

Thank you very much for your comments regarding our manuscript. We did our best to revise our paper according to your comments, suggestions, and comments and suggestions from another reviewer. Changes in the manuscript are made in red color.

Editing of the English language was done as well.

Nine new references were added to the revised manuscript; they are made in red.

Point-by-point responses to your questions and suggestions are included below.

Yours faithfully,

Pavel Mokrejš (corresponding author)

--------------------------------------------------

Comment and suggestion 1: The differences between using granular and liquid lipase are not sufficiently discussed. Please expand on this aspect, compare the results and highlight advantages and disadvantages of each.

Response: The comparison of granular and liquid lipases on degree of hydrolysis, as well as highlighting of pros and cons of both enzymes, is discussed in chapter “2.1. Degree of Fat Hydrolysis and Chemical Analysis“; please see 1st paragraph on page 6.

Comment and suggestion 2: Comment on the cost of the enzymatic hydrolysis, compared to other methods.

Response: These facts are newly included in the chapter “2.5. Proposal of Optimal Conditions for Enzymatic Hydrolysis of Deer Tallow and Contribution of the Study to the Praxis“.

Comment and suggestion 3: Explain why this specific range of the studied variables was chosen.

Response: This fact is explained in the chapter “3.2. Experimental Design and Statistical Analysis“.

Comment and suggestion 4: When comparing the results obtained with deer tallow with those reported for other sources (e.g. beef, poultry, salmon), discuss if the composition differences in the raw materials could explain the results. Also, expand on the different enzymes used in each case.

Response: The comparison of the results of hydrolysis based on the influence of composition differences (fatty acid profiles) of our deer tallow and different raw-material sources (References Nos. 37-40) is complicated to provide because processing conditions (type of enzyme, reaction time and temperature) are not the same in our case and the case of other authors. Nevertheless, we tried to discuss these facts in chapters “2.1. Degree of Fat Hydrolysis and Chemical Analysis“ and “2.2. Color“. Moreover, accurate comparisons between different enzymes (ours and those reported in the literature) were not entirely possible in some cases because some authors do not specify details of the type of enzyme used.

Comment and suggestion 5: Write caption of Figure 3 in a more impersonal style (e.g. "FTIR spectra of samples 1 and *10, etc." instead of "The FTIR spectra of sample 1 were treated, etc."). The provided information is adequate, but the structure of the phrase should be modified.

Response: The caption of Figure 3 was modified as suggested.

Comment and suggestion 6: The methods for processing the deer tallow should be explained in more detail.

Response: The processing steps of hydrolysis of deer tallow are explained in more detail, including essential processing facts.

Comment and suggestion 7: A list of abbreviations would be useful.

Response: New chapter “3.5. List of Abbreviations“ is added at the end of the revised paper.

Round 2

Reviewer 2 Report

Comments and Suggestions for Authors

The authors have responded to my comments and suggestions, so I propose publication.